

# Deriving Arctic 2 m air temperatures over snow and ice from satellite surface temperature measurements

Pia Nielsen-Englyst[1,2], Jacob L. Høyer[2], Kristine S. Madsen[2], Rasmus T. Tonboe[2], Gorm Dybkjær[2] and Sotirios Skarpalezos[2]

[1] Technical University of Denmark (DTU), DK-2800 Kongens Lyngby, Denmark
[2] Danish Meteorological Institute (DMI), DK-2100 Copenhagen Ø, Denmark

*Correspondence to*: Pia Nielsen-Englyst (pne@dmi.dk)

**Abstract.**

The Arctic region is responding heavily to climate change, and yet, the air temperature of ice covered areas in the Arctic is heavily under-sampled when it comes to in situ measurements, resulting in large uncertainties in existing weather- and reanalysis products. This paper presents a method for estimating daily mean clear sky 2 meter air temperatures (T2m) in the Arctic from satellite observations of skin temperature, using the Arctic and Antarctic ice Surface Temperatures from thermal Infrared (AASTI) satellite dataset, providing spatially detailed observations of the Arctic. The method is based on a linear regression model, which has been tuned against in situ observations to estimate daily mean T2m based on clear sky satellite ice surface skin temperatures. The daily satellite derived T2m product includes estimated uncertainties and covers clear sky snow and ice surfaces in the Arctic region during the period 2000-2009, provided on a 0.25 degree regular latitude-longitude grid. Comparisons with independent in situ measured T2m show average biases of 0.30°C and 0.35°C and average root mean square errors of 3.47°C and 3.20°C for land ice and sea ice, respectively. The associated uncertainties are verified to be very realistic for both land ice and sea ice, using in situ observations. The reconstruction provides a much better spatial coverage than the sparse in situ observations of T2m in the Arctic, is independent of numerical weather prediction model input and it therefore provides an important supplement to simulated air temperatures to be used for assimilation or global surface temperature reconstructions. A comparison between in situ T2m versus T2m derived from satellite and ERA-Interim/ERA5 estimates shows that the T2m derived from satellite observations validate similar or better than ERA-Interim/ERA5 in the Arctic.

## 1 Introduction

The Arctic climate is changing rapidly with surface temperatures rising faster than other regions of the world due to Arctic amplification (Graversen et al., 2008; IPCC, 2013; Pithan and Mauritsen, 2014; Richter-Menge et al., 2017). Meteorological measurements show that the 2000s were the warmest decade in Greenland since meteorological measurements started in the 1780s (Box et al., 2019; Cappelen, 2016; Masson-Delmotte et al., 2012).





The Arctic surface air temperature is one of the key climate indicators used to assess regional and global climate changes (Hansen et al., 2010; Pielke et al., 2007) and both model simulations and observations indicate that warming in the global climate is amplified at the northern high latitudes (e.g. Collins et al., 2013; Holland and Bitz, 2003; Overland et al., 2018). Traditionally, near surface air temperatures have been measured at the height of 1-2 m using automatic weather stations
(AWSs) or buoys (Hansen et al., 2010; Jones et al., 2012; Rayner, 2003; World Meteorological Organization, 2014). Extreme temperatures, winds and the remoteness of the Arctic make in situ observations in the Arctic temporally and spatially sparse (Reeves Eyre and Zeng, 2017), and challenging. In particular, it is difficult to achieve climate-quality temperature records for this region.

The key data sets used to assess the Arctic temperature changes are global gridded near surface air temperature datasets that
are derived by using in situ observations (Hansen et al., 2010; IPCC, 2013; Morice et al., 2012; Smith et al., 2008; Vose et al., 2012). These datasets typically have higher uncertainties in the Arctic region due to the limited availability of in situ observations (Cowtan and Way, 2014; Lenssen et al., 2019; Rapaić et al., 2015). In addition, global reanalysis products such as ERA-Interim (ERA-I) and ERA5 (Dee et al., 2011; Hersbach et al., 2020) are frequently used to study the changes in the Arctic and to force ocean and sea ice models. Despite the assimilation of in situ data in the global reanalysis models,
significant model differences have been reported for the Arctic (Davy and Outten, 2020; Delhasse et al., 2020; Lindsay et al., 2014; Wesslén et al., 2014) as well as large deviations from observations of T2m over Arctic sea ice (Wang et al., 2019).

Observations from polar orbiting satellites offer a very good supplement to the in situ observations through a high spatial and temporal coverage of the high latitudes and may improve the surface temperature products and the assessment of the Arctic climate changes. Daily near surface air temperatures derived from satellites temperature observations therefore have the
potential to increase the amount of information in the data sets and improve the quality of the climate records, as recognized in Merchant et al. (2013) and (Rayner et al., 2020).

However, two fundamental challenges exist when deriving a T2m product from infrared satellites that can supplement the existing products: The infrared sensors in the atmospheric window region of 10-12 micron wavelength measure the ice surface skin temperature ($IST_{skin}$) only in clear sky conditions, whereas the current global temperature products include the
near surface air temperature as measured by continuous AWSs and buoys. In addition, the surface skin temperature as observed by the satellite may differ considerably from the near surface air temperature. The largest differences occur during temperature inversions i.e. most commonly in winter, but also during melting conditions, where the surface temperature is fixed at the melting point. At other times e.g. during overcast conditions the skin and surface air temperature may be more or less the same (Nielsen-Englyst et al., 2019).
To benefit from the good coverage of satellite surface temperature data, we have explored the relationships between the surface air temperature and the satellite measurements with special attention to these challenges. Several studies have compared satellite retrieved $IST_{skin}$ and T2m from AWSs located on the Greenland Ice Sheet (GrIS; Dybkjær et al., 2012a; Hall et al., 2008, 2012; Koenig and Hall, 2010; Shuman et al., 2014) and over the Arctic sea ice (Dybkjær et al., 2012) and found temperature differences of which a significant part could be attributed to the temperature difference between T2m and



IST$_{skin}$. Previously, work has been done to investigate the relationship between the surface and near-surface air temperature over ice using in situ observations (Adolph et al., 2018; Hall et al., 2008, 2004; Hudson and Brandt, 2005; Nielsen-Englyst et al., 2019; Vihma et al., 2008). Nielsen-Englyst et al. (2019) found that on average T2m is 0.65-2.65°C higher than IST$_{skin}$ with variations depending on location of the measurement i.e. in the lower ablation zone, upper-middle ablation zone,

accumulation zone, seasonal snow cover and sea ice. The T2m-IST$_{skin}$ difference was found to vary with season with smallest differences around noon and early afternoon during spring, fall and summer during non-melting conditions. Furthermore, wind speed and cloud cover were identified as key parameters determining the T2m- IST$_{skin}$ difference.

Given the difficulties of operating equipment in the harsh Arctic conditions, the potential for using satellite IST$_{skin}$ to estimate T2m is large in this region. The greatest limitation of satellite-derived infrared surface temperatures is cloud cover.

Hence, a satellite-derived, clear-sky, surface temperature record can be significantly different from an all-sky surface temperature record (Koenig and Hall, 2010; Nielsen-Englyst et al., 2019).

This work, starting with Nielsen-Englyst et al. (2019), has been initiated to estimate clear sky T2m from satellite observations (whenever these are available) covering the snow and ice covered parts of the Arctic, in order to provide spatially-detailed observations for the areas unobserved by in situ stations and to supplement the in situ observations already

available.

A regression-based approach has been used to estimate daily T2m using satellite IST$_{skin}$ and a seasonal cycle function as predictors, based upon the work presented in Høyer et al. (2018). The derived product covers only days with none or limited clouds, where satellite skin temperature observations are available. However, for those days when the satellite derived T2m product is available, it provides an estimate of the daily averaged all-sky T2m, as it has been regressed towards in situ

measurements from both clear and cloudy conditions. In order to further facilitate the usage of the derived product in modelling and for monitoring purposes, each satellite retrieved T2m estimate comes with uncertainties. Similar efforts have been done to estimate clear sky near surface air temperatures (and corresponding uncertainties) over land, ocean and lakes using satellite observations to cover all surfaces of the Earth (Good, 2015; Good et al., 2017; Høyer et al., 2018). The previous work has mostly been done as a part of the European Union's Horizon2020 project EUSTACE (EU Surface

Temperatures for All Corners of Earth, 2015-2019, https://www.eustaceproject.org), with the overall aim to produce a globally complete gap-free daily near surface temperature analysis since 1850. It is outside the scope of this paper to produce a continuous gap-free daily near surface temperature analysis. However, within EUSTACE this has been done using a statistical model to combine the satellite derived clear sky near surface air temperatures (i.e. the product derived in this paper over ice and similar clear sky temperature products over land, ocean and lakes) and in situ observations, and their respective

uncertainty estimates (Morice et al., 2019; Rayner et al., 2020).

This paper is structured such that Sect. 2 describes the in situ data and the satellite data. Section 3 presents the method used to estimate clear-sky daily T2m and uncertainties. The resulting T2m dataset and its validation are presented in Sect. 4 and discussed in Sect. 5. Conclusions are given in Sect. 6.



## 2 Data

### 2.1 In Situ data

In situ observations of near surface air temperatures have been collected from weather stations, expeditions and campaigns covering ice and snow surfaces to assemble the DMI-EUSTACE database. The database includes quality controlled and

uniformly formatted temperature observations covering ice and snow surfaces, during 2000-2009 (Høyer et al., 2018). Over Arctic land ice/snow we use the Programme for Monitoring of the Greenland Ice Sheet (PROMICE) data provided by the Geological Survey of Denmark and Greenland (GEUS; Ahlstrøm et al., 2008), the Atmospheric Radiation Measurement (ARM) Program data from the North Slope of Alaska (Ackerman and Stokes, 2003; Stamnes et al., 1999), and the Greenland Climate Network data (GC-Net; Kindig, 2010; Shuman et al., 2001; Steffen and Box, 2001). Only PROMICE data from the

middle-upper ablation zone and accumulation zone have been used to ensure that data are only acquired over permanently snow or ice covered surfaces. Data on Arctic sea ice are primarily retrieved from the meteorological observation archive at the European Centre for Medium-Range Weather Forecasts (ECMWF) MARS data storage facility, providing 196 unique data series from drifting buoys. These sea ice data are supplemented with data from 10 U.S. Army Cold Regions Research Engineering Laboratory (CRREL) mass balance buoys (Perovich et al., 2016; Richter-Menge et al., 2006) and observations

from the research vessel, POLARSTERN, operated by the Alfred-Wegener-Institute, operating the sea ice covered parts of the Arctic Ocean (Knust, 2017). We also use air temperature measurements obtained from ice buoys deployed in the Fram Strait region within the framework of the Fram Strait Cyclones (FRAMZY) campaigns during the years 2002, 2007, and 2008 as well as air temperatures from the Arctic Climate System Study (ACSYS) campaign in 2003 (Brümmer et al., 2011b, 2011c, 2012b, 2012a). Finally, we use data from two ice buoy campaigns operated by the Meteorological Institute of the

University of Hamburg within the framework of the integrated EU research project DAMOCLES (Developing Arctic Modelling and Observing Capabilities for Long-term Environmental Studies; Brümmer et al., 2011a).

The different in situ types measure the air temperature at different heights that furthermore differ over time depending on the amount of snow fall, snow drift and snow melt. Here, we will refer to T2m for all observation types regardless of these variations. Nielsen-Englyst et al. (2019) showed small changes (<0.22°C) in T2m-$IST_{skin}$ differences when using only

observations within the measurement range of 1.90-2.10 m in height compared to using all measurements (ranging in measurement height from 0.3 m to 3 m). The accuracy of the air temperature sensors for all observation sites is approximated to 0.1°C (Hall et al., 2008; Høyer et al., 2017b). Few data sources provide both skin and air temperatures e.g. the PROMICE and ARM stations. The PROMICE skin temperatures have been calculated from up-welling longwave radiation, measured by Kipp & Zonen CNR1 or CNR4 radiometer, assuming a surface longwave emissivity of 0.97 (van As, 2011). All in situ

data have been screened for spikes and other unrealistic data artefacts by visual inspection. Afterwards, the in situ observations have been averaged to daily temperatures using all available observations. Figure 1 shows the number of daily averaged in situ observations each year during 2000-2009 of $IST_{skin}$ and T2m over Arctic land ice and sea ice. In total 65,810





observations with daily T2m and 7,057 observations with daily $IST_{skin}$ are available over land ice. See Table 1 for more information on the in situ observations used in this study.

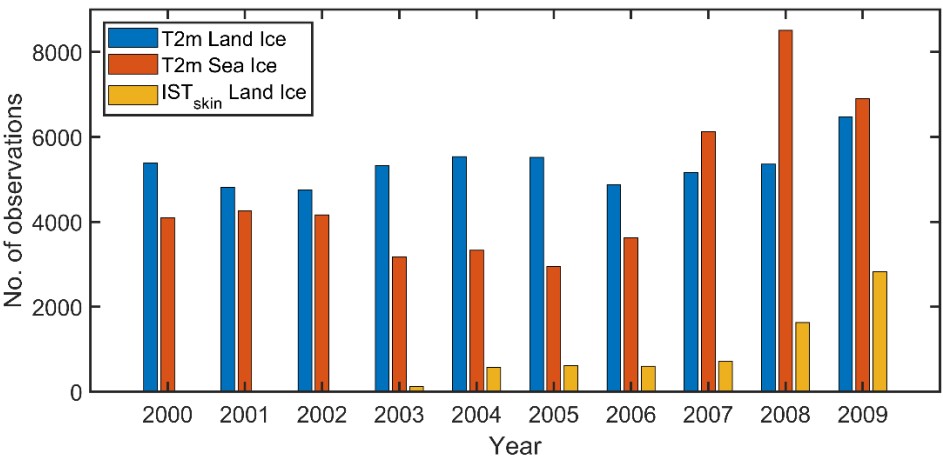

5 **Figure 1: Total number of daily averaged in situ observations of T2m and $IST_{skin}$ over Arctic land ice and sea ice per year covering the period 2000-2009.**

**Table 1: Overview of in situ observations used in this study, covering 2000-2009.**

|  | No. of sites, (AWS, buoys or ships) | No. of days with observations | Surface Type | Observation Type | Temperature measurements |
|---|---|---|---|---|---|
| ACSYS | 7 | 280 | Sea ice | Buoy | T2m |
| ARM | 2 | 2,846 | Land snow | AWS | T2m, $IST_{skin}$ |
| CRREL | 10 | 1,031 | Sea ice | Buoy | T2m |
| DAMOCLES | 25 | 2,160 | Sea ice | Buoy | T2m |
| ECMWF | 196 | 27,235 | Sea ice | Buoy | T2m |
| FRAMZY | 11 | 251 | Sea ice | Buoy | T2m |
| GC-NET | 15 | 29,133 | Land ice | AWS | T2m |
| POLARSTERN | 1 | 189 | Sea ice | Ship | T2m |
| PROMICE | 8 | 2,685 | Land ice | AWS | T2m, $IST_{skin}$ |

10 **2.2 Satellite data**

The satellite data used in this study is from the Arctic and Antarctic Ice Surface Temperatures from thermal Infrared satellite sensors (AASTI; Dybkjaer et al., 2018; Dybkjær et al., 2014; Høyer et al., 2019) data set, covering high latitude seas, sea ice,



and ice cap clear sky surface temperatures based on satellite infrared measurements from the CLARA-A1 data set compiled by EUMETSAT's Climate Monitoring, Satellite Application Facility (Karlsson et al., 2013). The data set is based on one of the longest existing satellite records from the Advanced Very High Resolution Radiometer (AVHRR) instruments on board a long series of NOAA satellites. AASTI contains swath based (i.e. Level 2; L2) ice surface skin temperature (IST$_{skin\_L2}$) data

processed and error corrected on the original Global Area Coverage (GAC) grid. The first version of the AASTI product, which is used in this study, is available from 2000 to 2009 in the original projection and resolution (L2), i.e. ~0.05 arc degree resolution and multiple daily coverage. Since 2000, seven different AVHRR instruments have been orbiting the globe, each 14 times per day, and thus providing approximately bi-hourly coverage of the Polar Regions (Figure 2). The number of operational satellites has increased from 2 to 6 from 2000 to 2009. The IST algorithm used to generate the AASTI data set is

based on thermal infrared brightness temperatures of AVHRR channel 4 (centre wavelength at ~11 microns) and 5 (centre wavelength at ~12 microns), and the satellite zenith angle. The algorithm is a split window algorithm, working within three temperature domains for each individual satellite (Key et al., 1997). The retrieval calibration of each domain has been done by relating modelled surface temperatures with modelled top-of-atmosphere brightness temperatures, determined by a radiative transfer model (Dybkjær et al., 2014). Cloud masking has been performed using the Polar Platform System (PPS)

processing cloud processing software (Dybbroe et al., 2005a, 2005b).

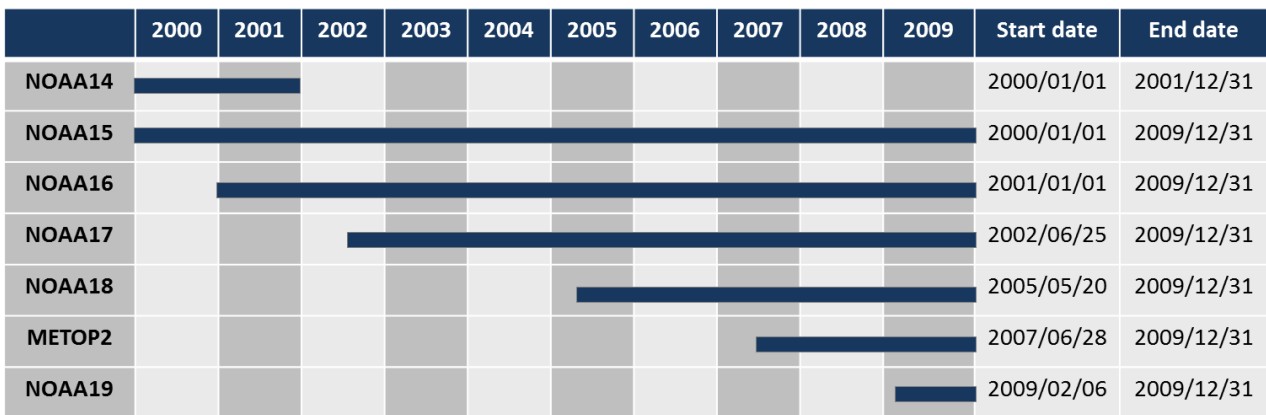

**Figure 2: NOAA and Metop satellites carrying the AVHRR sensor, used for AASTI version 1.**

As discussed in Merchant et al. (2017), satellite-based climate data records should include uncertainty estimates. The AASTI

IST$_{skin\_L2}$ data come with uncertainties divided into three independent uncertainty components, each with different characteristics: the random uncertainty ($\mu_{rnd\_L2}$), a locally systematic uncertainty ($\mu_{local\_L2}$) and a large-scale systematic ("global") uncertainty ($\mu_{glob\_L2}$). These three components have been chosen since they behave differently when aggregating the observations in time or space (see Sect. 3.2). This uncertainty methodology has been developed within the sea surface temperature (SST) community (Bulgin et al., 2016; Rayner et al., 2015) and will be followed here. The total uncertainty on

the IST$_{skin\_L2}$, $\mu_{total\_L2}$, is calculated by summing each component in quadrature (i.e., square root of sum of squares).



Excluding the cloud mask uncertainty, grid-cell systematic uncertainties ($\mu_{glob\_L2}$) are set to a fixed value of 0.1°C to represent systematic uncertainties in the forward models (see e.g. Merchant et al., 1999; Merchant and Le Borgne, 2004). The AASTI $IST_{skin\_L2}$ data also come with a quality level (QL) from 1 (bad data) to 5 (best quality), with the addition of level 0 (no data) (GHRSST Science Team, 2010).

Here, we have aggregated the AASTI $IST_{skin\_L2}$ observations into 3 hourly and daily, gridded Level 3 (L3) averages ($IST_{skin\_L3}$) of $IST_{skin\_L2}$ on a fixed 0.25 by 0.25 degrees regular geographical grid. The $IST_{skin\_L3}$ is calculated by averaging all available $IST_{skin\_L2}$ observations with a quality flag of 4 (good) or 5 (best) for a given date and within the 0.25 degree bin. This has been done to facilitate the development of the relationship model and to ease the user uptake. The data in the daily aggregated files contain mean surface temperature observations from 00 to 24 hours local solar time, 3-hourly bin averages

of surface temperatures and also the number of observations in the eight time bins during each day. The 3-hourly numbers of observations are used for estimating the satellite sampling throughout the day, and the 3-hourly temperature data to gain confidence in the daily cycle estimates (see quality checks below). Figure 3 shows the mean number of observations per day in each of the eight time intervals given in local time for the Arctic region. The variation in the coverage throughout the day is a combined effect of the satellite overpassing, performance of the cloud screening algorithm, and the cloud free conditions

during the day. In addition, the fixed 0.25 degrees regular geographical grid results in a decreasing L3 bin area when approaching the North Pole. The maximum in satellite coverage is generally seen around 80°N with a minimum at the North Pole. Cloud free conditions over the GrIS are primarily observed around noon and early afternoon.

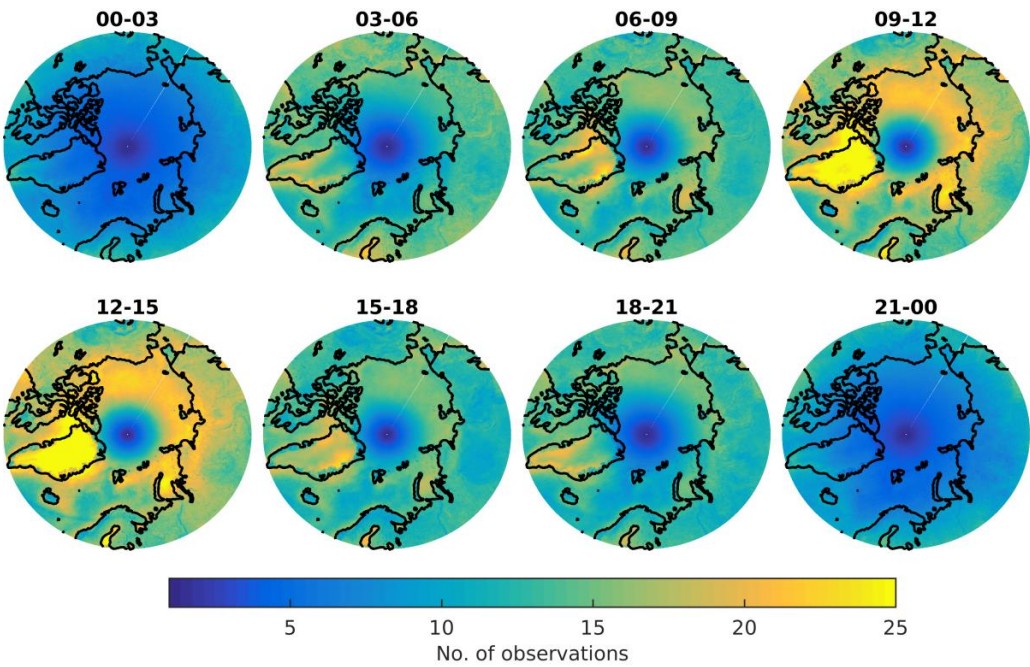

**Figure 3: Mean number of observations per day in the L3 bins for each of the eight local solar time intervals, averaged for the**
**period 2000-2009.**



In order to best resolve the diurnal cycle with satellite information, we require data during both night (between 18 and 6 local solar time) and day (between 6 and 18 local solar time) in order to calculate $IST_{skin\_L3}$. A few more checks have been set up in order to minimize the temporal sampling errors and the effects of undetected clouds and outliers. Following Høyer et al. (2018), the $IST_{skin\_L3}$ is discarded if one of the following criteria is met:

- $IST_{skin\_L3}$ exceeds +5°C, indicating clear melting conditions or obviously wrong observations.
- The standard deviation of satellite $IST_{skin\_L2}$ during one day exceeds 7.07°C, corresponding to a sinusoidal daily cycle with a difference between day and night of 20°C.
- The difference between $IST_{skin\_L3}$ and the average of all available 3 h bin averages exceeds 10°C.
- $IST_{skin\_L3}$ is more than 10°C colder than the corresponding average of up to 24 neighbouring cloud free observations
(in a 5 by 5 grid cell square) with the same surface type.

The criteria above have been derived from analysis and inspection of the satellite data and with considerations to the results presented in Nielsen-Englyst et al. (2019). The satellite-derived $IST_{skin\_L3}$ has seasonal differences in daily variability, with largest standard deviations during summer in Greenland and during winter for sea ice, where the freeze-up of sea ice causes higher variability along the sea ice margin (Fig. 4). The main uncertainty components of the $IST_{skin\_L3}$ estimates are

erroneous cloud screening and the spatial variance of snow and ice surface emissivity, which are not accounted for in the retrieval algorithm. The presence of non-detected clouds will contribute to increased standard deviations and usually a cold $IST_{skin\_L3}$ bias, since the cloud tops and other atmospheric constituents generally are colder than the surface (Dybkjær et al., 2012).

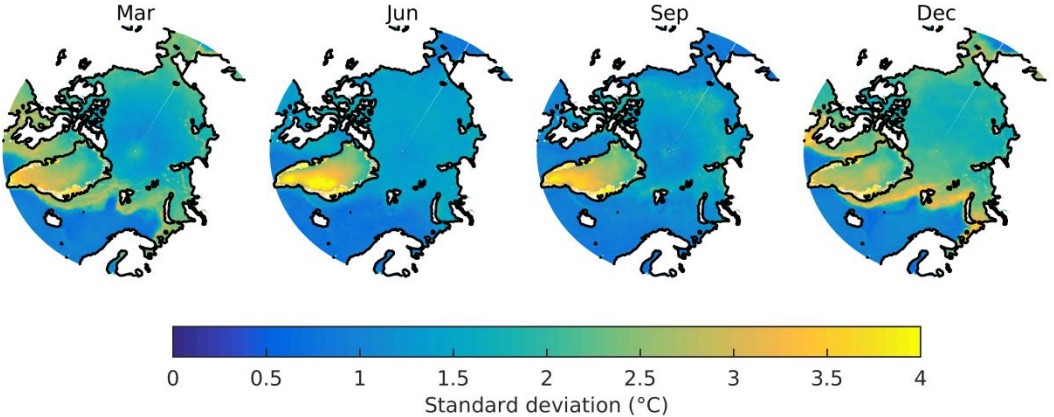

**Figure 4: Standard deviations of daily satellite surface temperature observations for March, June, September and December, averaged for the years 2000-2009 (°C).**

Additional satellite versus in situ differences arise when comparing satellite observations with pointwise ground measurements due to different spatial and temporal characteristics. To assess the magnitude of these effects, the $IST_{skin\_L3}$ data have been validated against in situ land ice temperatures from the PROMICE and ARM stations. Table 2 shows the

validation results of $IST_{skin\_L3}$ against in situ skin temperatures ($IST_{skin\_insitu}$) and in situ 2 meter air temperatures ($T2m_{insitu}$), respectively. The maximum matchup distance is 14.6 km and the average distance is 8.1 km, considering the AWSs in Table

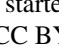



2. The topography mask included in the HIRHAM5 regional climate model (see e.g. Langen et al. 2015) has been used to calculate the differences in elevation (Δh) between the in situ stations and corresponding satellite pixels. There is no clear correlation between the large biases and large elevation differences from this table, but the elevation effects are contributing to the spatial sampling error. The spatial and temporal sampling errors contribute to the overall uncertainty, but effects from erroneous cloud screening, algorithm simplifications, and uncertainties in the in situ observations are also included in the results. Previous studies find that erroneous cloud screening (undetected clouds) is one of the main reasons for the cold biases observed when comparing satellite observed IST with in situ measurements (Hall et al., 2004, 2012; Koenig and Hall, 2010; Østby et al., 2014; Westermann et al., 2012). Another important contribution is the effect of comparing clear sky satellite observations with all-sky in situ observations, as discussed in Nielsen-Englyst et al. (2019). In general, $IST_{skin\_L3}$ correlates better with $T2m_{insitu}$ than with the $IST_{skin\_insitu}$. Moreover, the $IST_{skin\_L3}$-$T2m_{InSitu}$ difference shows smaller standard deviations than $IST_{skin\_L3}$-$IST_{skin\_insitu}$. However, as expected the biases and root mean squared differences (RMS) are larger for the $IST_{skin\_L3}$-$T2m_{insitu}$ differences than for the $IST_{skin\_L3}$-$IST_{skin\_insitu}$ differences. The reason is that the radiometric surface skin temperature can be significant different from the surface air temperature measurements (Adolph et al., 2018; Hall et al., 2008; Hudson and Brandt, 2005; Nielsen-Englyst et al., 2019; Vihma et al., 2008). On average, the skin temperature is colder than the air temperature (Nielsen-Englyst et al., 2019), resulting in even more negative biases, when the $IST_{skin\_L3}$ is compared to in situ measured T2m, instead of in situ skin temperatures. The generally high correlations are dominated by the synoptic (2-5 days) and seasonal variations, which are pronounced in both IST and T2m.

**Table 2. Validation of daily AASTI v.1 Level 3 IST ($IST_{skin\_L3}$) against in situ $IST_{skin}$ ($IST_{skin\_insitu}$) and T2m observations ($T2m_{insitu}$). N: number of matchups, Corr: correlation, Std: standard deviation, RMS: root mean square difference, d is the matchup distance and Δh is the difference in elevation (AWS – Satellite).**

| Station | N | $IST_{skin\_L3}$ - $IST_{skin\_insitu}$ | | | | $IST_{skin\_L3}$ – $T2m_{insitu}$ | | | | d (km) | Δh (m) |
|---|---|---|---|---|---|---|---|---|---|---|---|
| | | Corr | Bias | Std | RMS | Corr | Bias | Std | RMS | | |
| ARM_Atq | 1235 | 93.8 | -2.47 | 3.69 | 4.44 | 93.7 | -3.17 | 3.69 | 4.87 | 10.8 | - |
| ARM_Bar | 1594 | 94.1 | -0.73 | 4.30 | 4.36 | 94.6 | -1.14 | 4.02 | 3.86 | 6.1 | - |
| PROMICE KAN-M | 422 | 93.9 | -3.65 | 3.37 | 4.96 | 94.6 | -4.56 | 3.14 | 5.53 | 7.6 | 15 |
| PROMICE KAN-U | 239 | 93.9 | -1.75 | 3.32 | 3.75 | 94.4 | -3.39 | 3.17 | 4.64 | 14.6 | 21 |
| PROMICE KPC-U | 488 | 97.6 | -1.31 | 2.62 | 2.92 | 98.2 | -3.20 | 2.27 | 3.92 | 5.1 | 29 |
| PROMICE NUK-U | 296 | 77.7 | -4.09 | 5.00 | 6.45 | 84.7 | -7.19 | 4.01 | 8.23 | 14.4 | 64 |
| PROMICE QAS-U | 407 | 83.9 | -1.65 | 4.20 | 4.51 | 86.3 | -3.70 | 3.75 | 5.27 | 6.5 | 197 |
| PROMICE SCO-U | 403 | 91.5 | -4.60 | 4.25 | 6.26 | 93.7 | -7.55 | 3.75 | 8.43 | 4.2 | 20 |
| PROMICE TAS-U | 386 | 67.5 | -1.03 | 5.43 | 5.52 | 79.5 | -3.61 | 4.39 | 5.68 | 8.4 | 214 |
| PROMICE UPE-U | 125 | 88.2 | -3.13 | 3.88 | 4.97 | 90.0 | -5.49 | 3.50 | 6.50 | 3.0 | 110 |
| All data | 5595 | 92.9 | -2.03 | 4.24 | 4.70 | 93.2 | -3.36 | 4.12 | 5.32 | 8.1 | 83.8 |





## 3 Methods

### 3.1 Regression model

Nielsen-Englyst et al. (2019) analysed a large number of in situ stations with simultaneous T2m and $IST_{skin}$ observations and showed that empirical relationships exist between T2m and $IST_{skin}$. It was also shown, however, that the relationships varied

for different regions. Based upon these results, it was decided to use a simple regression based method in this paper to derive the daily mean T2m from the satellite $IST_{skin\_L3}$ observations. Separate regression models have been derived for land ice and sea ice.

To test different types of regression models, the $IST_{skin\_L3}$ data have been matched up with in situ observations for each day (Høyer et al., 2018). This is done by requiring a distance to nearest in situ site of less than 15 km. The average matchup

distance is 8.6 km and 7.2 km for land and sea ice, respectively, which means that all in situ sites are located within the area of the satellite pixel. The corresponding mean elevation difference is 30 m (while the absolute mean elevation difference is 45 m) and is calculated using the topography mask included in HIRHAM5 (Langen et al., 2015) for the 23 GrIS AWSs. Considering the 23 AWSs, four of them (GC-net JAR1, TAS_U, QAS_U and UPE_U) have corresponding elevation differences above 100 m. In the Section 5, the effect of these AWSs has been discussed. All in situ observations, described in

Sect. 2.1., have been matched up with $IST_{skin\_L3}$ data, resulting in a total number of daily matchups of 65,810 from 275 different observation sites (see Table 1). These have been divided into two subsets: one for training and one for validation of the different regression models for land- and sea ice, respectively. This has been done while ensuring similar coverage of training and validation data over the two domains, which is shown in Fig. 5. The result is that 40% (13,792 matchups) are used for testing the regression models (and generating the regression coefficients) and the remaining 60% (20,872 matchups)

are left for validation of the regression models over land ice. Over sea ice 48% (15,035 matchups) are used for testing and 52% (16,111 matchups) are left for validation.

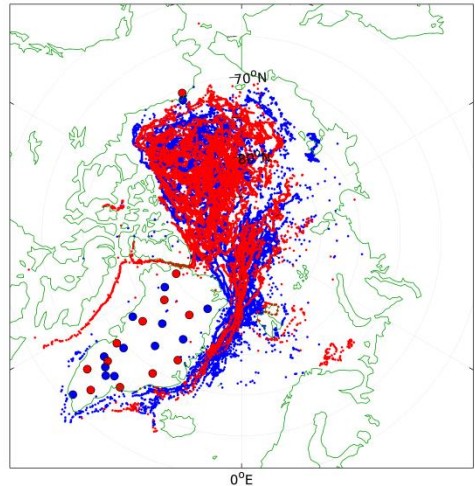

**Figure 5: Positions of matchups on sea ice and land ice (red: training, blue: validation)**





The regression model is based on multiple linear regression analysis using least squares (Menke, 1989). The multiple linear regression analysis equations can be written in matrix form,

$$\boldsymbol{d}^{obs} = \mathbf{G}\boldsymbol{m} + \boldsymbol{e} \qquad (4)$$

$$\boldsymbol{d}^{pre} = \mathbf{G}\boldsymbol{m}, \qquad (5)$$

where $\boldsymbol{d}^{obs}$ and $\boldsymbol{d}^{pre}$ are vectors containing the observed and modelled in situ air temperatures, respectively, $\mathbf{G}$ is a matrix containing the various predictors, $\boldsymbol{m}$ is a vector containing regression coefficients, and $\boldsymbol{e}$ is the fitting error.

The regression coefficients are found using damped least squares (Menke, 1989). The least squares method is used since the problem is generally over-determined, and the damping is added to limit effects of noisy data. The regression coefficients are thus given as:

$$\mathbf{G}^{-g} = (\mathbf{G}^{\mathbf{T}}\mathbf{G} + \varepsilon^2 \mathbf{I})^{-1}\mathbf{G}^{\mathbf{T}} \qquad (6)$$

$$\boldsymbol{m} = \mathbf{G}^{-g}\boldsymbol{d}^{obs}, \qquad (7)$$

where $\mathbf{G}^{-g}$ is called the generalized inverse, ε is a damping factor and $\mathbf{I}$ is an identity matrix (with ones in the diagonal and zeros elsewhere). The superscript operator $\mathbf{T}$ denotes transposing and -1 denotes inversion. We have tested a range of damping factors to assess the relation to the error coefficients. A damping factor of 0.2 was chosen to avoid overfitting noise in the data, while keeping the error coefficients low.

The choice of predictors is based on current knowledge of the parameters that influence the relationship between $IST_{skin}$ and $T2m_{insitu}$ (Nielsen-Englyst et al., 2019), limited by the available satellite data. Nielsen-Englyst et al. (2019) showed that the T2m-Tskin difference varies over the season with smallest differences during spring, fall and summer in non-melting conditions. For that reason, we have also tested the effect of including a seasonal cycle as predictor. A total of 5 regression models with different predictors have been tested (Høyer et al., 2018):

$\hat{I}ST_{skin}$: 
$$T2m_{sat} = \alpha_0 + \alpha_1 IST_{skin\_L3} \qquad (8)$$

$\hat{I}ST_{skin}SWd$: 
$$T2m_{sat} = \alpha_0 + \alpha_1 IST_{skin\_L3} + \alpha_2 SWd \qquad (9)$$

$\hat{I}ST_{skin}WS$: 
$$T2m_{sat} = \alpha_0 + \alpha_1 IST_{skin\_L3} + \alpha_2 WS \qquad (10)$$

$\hat{I}ST_{skin}Lat$: 
$$T2m_{sat} = \alpha_0 + \alpha_1 IST_{skin\_L3} + \alpha_2 Lat \qquad (11)$$

$\hat{I}ST_{skin}Season$: 
$$T2m_{sat} = \alpha_0 + \alpha_1 IST_{skin\_L3} + \alpha_2 \cos\left(\frac{t \cdot 2\pi}{1\,yr}\right) + \alpha_3 \sin\left(\frac{t \cdot 2\pi}{1\,yr}\right) \qquad (12)$$

The regression model in Eq. (8) is limited to an offset and a scaling of $IST_{skin\_L3}$, where the latter term accounts for the synoptic and seasonal variations, which are the dominating factors in both the IST and T2m variability. This part is thus



included in all regression models tested. The other regression models also have a third predictor, which is included to examine how to best represent the residual variations in the T2m-IST difference. The model in Eq. (9) uses theoretical shortwave radiation, Eq. (10) uses the wind forcing (from ERA-I and ERA5, respectively), Eq. (11) uses latitude variation, and Eq. (12) uses a seasonal variation. In the regression model in Eq. (12), the seasonal variation is assumed to be the shape

of a cosine function, $A \cdot \cos\left(\frac{t \cdot 2\pi}{1\,yr} - \varphi\right)$, where A is the amplitude, $\varphi$ is the phase and t is time. Since $\cos(x_1 - x_2) = \cos(x_1)\cos(x_2) + \sin(x_1)\sin(x_2)$, the seasonal cycle can be rewritten to the form in Eq. (12) with $A = \sqrt{\alpha_2{}^2 + \alpha_3{}^2}$ and $\varphi = \arctan\left(\frac{\alpha_3}{\alpha_2}\right)$.

The training data have been used to calculate the regression coefficients for each regression model covering land ice and sea ice, respectively. The performance of each regression model has been investigated using the training data and the results are

shown in Table 3. The best performance is found by using the regression model where T2m$_{sat}$ is predicted from IST$_{skin\_L3}$ combined with a seasonal variation (ÎST$_{skin}$Season). This model predicts T2m$_{sat}$ better compared to the other regression models, with correlations above 96% and RMS values of 3.25-3.28°C against training data for both surface types (Table 3). In the following, we will use the regression model given in Eq. (12) with the seasonal term included and with separate regression coefficients for land ice and sea ice. The values are shown in Table 4. The phase corresponds to a maximum the

19[th] January and 12[th] February for land ice and sea ice, respectively. This is in agreement with Nielsen-Englyst et al. (2019) who found the strongest clear-sky inversion during the winter months (Dec-Feb) for all sites included in the analysis except from the ones located in the lower ablation zone (not included here), where pronounced surface melt takes place for long periods of time.

**Table 3: Statistics on the relation between observed and modelled temperatures for the training data. N: number of matchups used for testing, Corr: correlation, RMS: root mean square difference. Since, the training data are used for the regression, the bias is zero and thus the standard deviation equals RMS.**

|  |  | N | Corr (%) | RMS (°C) |
|---|---|---|---|---|
| Land ice | ÎST$_{skin}$ | 13792 | 95.7 | 3.51 |
|  | ÎST$_{skin}$SWd | 13792 | 96.2 | 3.28 |
|  | ÎST$_{skin}$WS$_{ERA-I}$ | 13792 | 95.8 | 3.47 |
|  | ÎST$_{skin}$WS$_{ERA5}$ | 13792 | 95.9 | 3.42 |
|  | ÎST$_{skin}$Lat | 13792 | 95.8 | 3.48 |
|  | ÎST$_{skin}$Season | 13792 | 96.3 | 3.28 |
| Sea ice | ÎST$_{skin}$ | 15035 | 96.0 | 3.32 |
|  | ÎST$_{skin}$SWd | 15035 | 96.0 | 3.32 |
|  | ÎST$_{skin}$WS$_{ERA-I}$ | 15035 | 96.0 | 3.32 |
|  | ÎST$_{skin}$WS$_{ERA5}$ | 15035 | 96.0 | 3.32 |





| | | | | |
|---|---|---|---|---|
| $\hat{IST}_{skin}Lat$ | 15035 | 96.1 | 3.28 | |
| $\hat{IST}_{skin}Season$ | 15035 | 96.2 | 3.25 | |

**Table 4: Model regression coefficients for $\ddot{I}ST_{skin}Season$.**

| | Offset, $\alpha_0$ (°C) | $IST_{skin\_L3}$ factor, $\alpha_1$ | Amplitude, A | Phase, $\varphi$ |
|---|---|---|---|---|
| Land ice | 4.20 | 1.06 | 2.26 | -0.33 |
| Sea ice | 1.46 | 0.89 | 1.83 | -0.75 |

## 3.2 Uncertainty estimates for T2m$_{sat}$

Uncertainty estimates on the derived T2m$_{sat}$ are crucial to facilitate the usage of the data set in modelling and for monitoring purposes. The uncertainty estimates of the satellite-derived T2m$_{sat}$ data follow the approach in Bulgin et al. (2016) and Rayner et al. (2015), which has also been used for the AASTI data. The uncertainty on a single T2m$_{sat}$ estimate is divided into random, locally correlated and systematic uncertainty components, with the total uncertainty $\mu_{total\_t2m}$ given as the square root of the sum of the three squared components:

$$\mu_{total\_T2m} = \sqrt{\mu_{rnd\_T2m}{}^2 + \mu_{local\_T2m}{}^2 + \mu_{glob\_T2m}{}^2}$$

The random uncertainty component for the T2m$_{sat}$ belonging to a particular grid cell at a particular point in time is found by propagating the AASTI IST$_{skin\_L3}$ random uncertainty through the regression model:

$$\mu_{rnd\_T2m} = \sqrt{\left(\alpha_1\mu_{rnd\_L3}\right)^2},$$

with $\mu_{rnd\_L3}$ given as the aggregated $\mu_{rnd\_L2}$:

$$\mu_{rnd\_L3} = \frac{\mu_{rnd\_L2}}{\sqrt{N}},$$

where $N$ is the number of observations for each bin in the aggregation from L2 to L3. The $\sqrt{N}$ reduction applies because the random uncertainty of each L2 data point that goes into the L3 calculation is by definition independent from the other.

The L3 global uncertainty component does not average out in any aggregation and is thus transferred directly from the L2 uncertainty estimate and has been multiplied by $\alpha_1$ to make up $\mu_{glob\_T2m}$:

$$\mu_{glob\_T2m} = \alpha_1\mu_{glob\_L3} = \alpha_1 \cdot 0.1°C$$

The $\mu_{local\_T2m}$ contains both the local uncertainty component of L2, a sampling error $\mu_{lsamp\_L3}$ related to sampling errors in space and time due to the aggregation, a relationship error, cloud mask uncertainty etc. When aggregating from L2 to daily L3, additional sources of uncertainty enter through the gridding process as IST$_{skin\_L3}$ can only be retrieved for clear-sky pixels. This introduces a temporal and spatial sampling uncertainty. If all our satellite observations were obtained during all-



sky conditions we assume that the high polar temporal coverage is such that the temporal sampling uncertainty in the L3 files can be set to zero. However, this is not the case and using only clear-sky observations generally leads to a clear-sky bias in averaged $IST_{skin}$ satellite observations when compared to in situ observations (Hall et al., 2012; Nielsen-Englyst et al., 2019; Rasmussen et al., 2018). The relationship error represents the standard deviation of the residuals calculated at in situ stations,

where both skin and air temperatures are available, i.e. $T2m_{sat}$-$T2m_{insitu}$. Estimating all the different components that make up the $\mu_{local\_T2m}$ is a very challenging task and is out of the scope of this paper. Instead, we estimate the $\mu_{local\_T2m}$ component using a simple regression model fitted to the satellite derived T2m and in situ T2m differences. Separate models have been chosen for the land ice and sea ice, due to the differences in the error characteristics. The variables to include in the uncertainty regression models have been chosen from a careful examination of the matchup data set. For land ice and sea ice

the most relevant variables were the $IST_{skin\_L3}$ itself and the number of 3 h time bins with observations in the L3, $N_{bins}$.

For land ice the regression model for $\mu_{local\_T2m}$ is given as following:

$$\mu_{local\_T2m\_landice} = \beta_0 + \beta_1 IST_{skin\_L3} + \beta_2 N_{bins} ,$$

while the regression model for sea ice is given as:

$$\mu_{local\_T2m\_seaice} = \gamma_0 + \gamma_1 IST_{skinL3} + \gamma_2 IST_{skinL3}^2 + \gamma_3 N_{bins} .$$

The coefficients have been determined by fitting to the $T2m_{sat}$-$T2m_{insitu}$ standard deviations calculated for the training data with $IST_{skin\_L3}$ bin intervals of 2°C and $N_{bins}$ interval of 1. The $\mu_{rnd\_T2m}$ and $\mu_{glob\_T2m}$ components have been removed from the standard deviations in each bin as well as an assumed in situ uncertainty of 0.1°C and an average sampling uncertainty of 0.5°C (Høyer et al., 2017a; Reeves Eyre and Zeng, 2017) before fitting the regression models. The optimal regression coefficients for each domain are listed in Table 5.

**Table 5: Uncertainty model regression coefficients**

| | | | | |
|---|---|---|---|---|
| Land ice | $\beta_0$ = 3.82°C | $\beta_1$ = -0.24 | $\beta_2$ = -0.03 | |
| Sea ice | $\gamma_0$ = 2.01°C | $\gamma_1$ = -0.06 | $\gamma_2$ = -0.12 | $\gamma_3$ = -0.001 |

**4 Results**

In Sect. 3.1, we selected the best (Eq. 12) of the 5 different algorithms and the derived coefficients (Table 3 and 4) to retrieve T2m from satellite surface temperature estimates. The dataset consists of daily estimates of mean air temperature on a 0.25

degree regular latitude-longitude grid, during the period 2000-2009 (Høyer et al., 2018; Kennedy et al., 2019). Days with clouds and few clear sky observations (as explained in Section 2.2) are not included in the dataset. However, for those days when the satellite derived T2m product is available, it provides an estimate of the daily averaged all-sky T2m (see Sect. 5). Each temperature estimate is associated with three components of uncertainty: random uncertainties on the 0.25 degree daily scale, synoptic scale correlated uncertainty and globally correlated uncertainty excluding uncertainties related to the masking





of clouds. The three types of uncertainties are also gathered in a total uncertainty estimate. The land ice temperatures have been calculated for grid cells categorized as ice shelf by ETOPO1, averaged to the 0.25 degree grid (Amante and Eakins, 2009). Sea ice temperatures have been calculated for grid cells with sea ice concentrations above 30%, according to OSISAF (Tonboe et al., 2016).

5  An evaluation of the product and the $T2m_{sat}$ regression model performance has been carried out by a comparison to the independent in situ data (i.e. the validation subset described in Sect. 3.1). Figure 6 shows an example of the daily near surface air temperature coverage (from Jan 1$^{st}$, 2008). Circles are in situ T2m measurements from coincidence independent AWSs and buoys and there seems to be quite good agreement between these and $T2m_{sat}$ during this specific day. The overall model performance, when compared to all independent AWS and buoy observations, is summarized in Table 6. The satellite

10  derived air temperatures are about 0.3°C warmer than measured in situ air temperature for both land ice and sea ice. The correlations are above 95% for both surface types and the RMS is 3.47°C and 3.20°C for land and sea ice, respectively. Note that the uncertainty of the in situ data is also included in these RMS values.

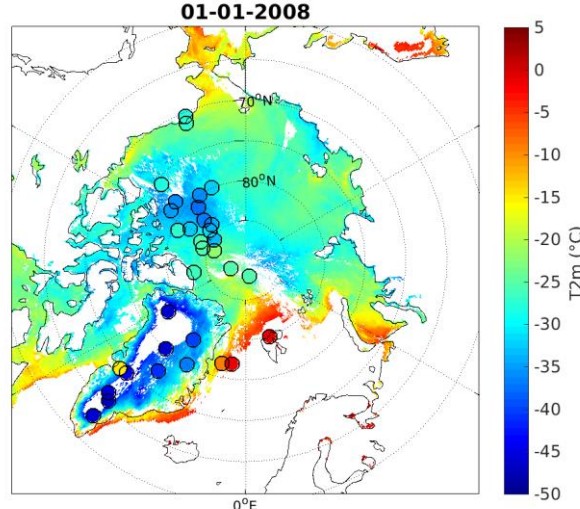

**Figure 6: Daily mean air surface temperature over land ice and sea ice from January 1, 2008. Circles show in situ measurements.**

**Table 6: Statistics on the relation between satellite-derived and in situ measured temperatures for comparison with independent validation data. N: number of matchups used for validation, Corr: correlation, bias: T2m$_{sat}$ − T2m$_{insitu}$ difference, Std: standard deviation, RMS: root mean square difference.**

|          | N     | Corr (%) | bias (°C) | Std (°C) | RMS (°C) |
|----------|-------|----------|-----------|----------|----------|
| Land ice | 20872 | 95.5     | 0.30      | 3.45     | 3.47     |
| Sea ice  | 16111 | 96.5     | 0.35      | 3.18     | 3.20     |

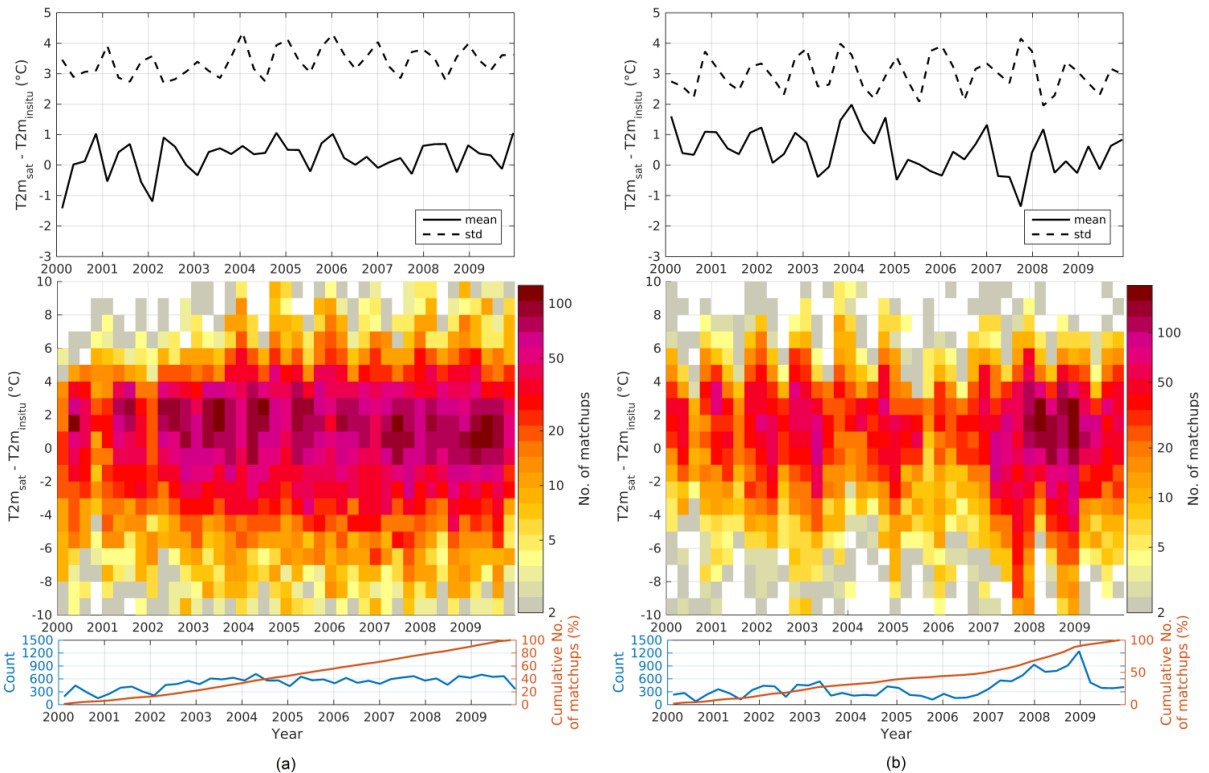

**Figure 7: Estimated T2m minus observed T2m (bin size of 1°C) for the full time period (bin size of 90 days) for (a) land ice and (b) sea ice. The dashed lines are standard deviations while the solid lines are bias in the upper figure. The surface plots in the middle figures show the number of matchups in each bin, while the bottom plots show the number of matchups (blue) and the cumulative percentage of matchups (red) in each time bin.**

Figure 7 shows the seasonal averaged independent validation statistics for land ice and sea ice, respectively. For both land ice and sea ice there are a seasonal dependency in standard deviation with largest values during winter and smallest during summer. This is probably explained by a better cloud screening performance during sunlit periods (Karlsson and Dybbroe, 2010). No significant seasonal cycle is seen in the mean bias, except for the sea ice region during 2000-2004, where there is a tendency to a warm bias during December and January.



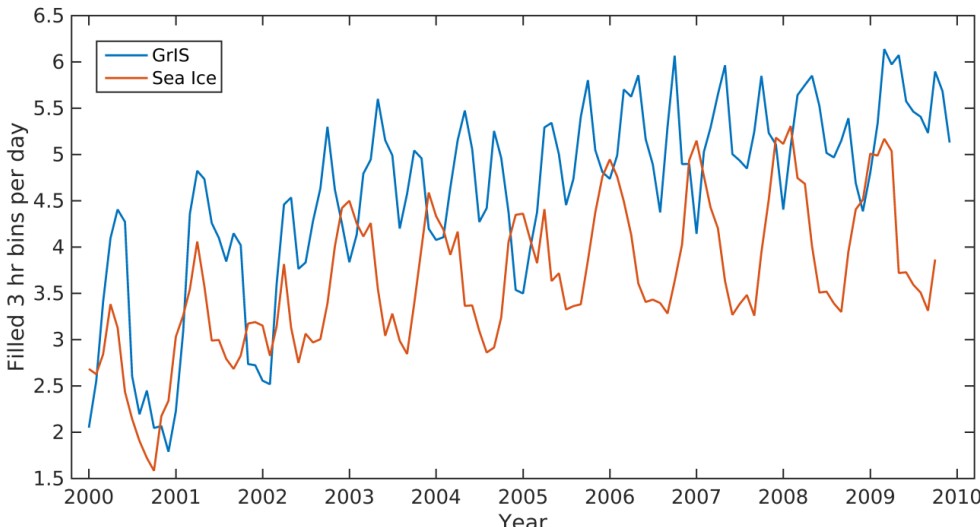

**Figure 8: Average number of filled 3 h bins per day for the Greenland Ice Sheet and the Arctic sea ice, respectively.**

As more satellite observations have become available over the time period a better coverage of the surface temperature is expected over time. Figure 8 shows the average number of filled 3 h bins per day for the GrIS and Arctic sea ice, 2000-2009.

Both surface types show an increase in filled 3 h bins over time, with large seasonal variations. In most years sea ice has 1-1.5 filled bins per day more during winter than summer, due to a more extensive cloud cover over sea ice during summer. The GrIS typically has fewer filled bins per day during winter and summer, than spring and fall, which is also explained by differences in cloud coverage. Note that the increase in the average number of filled 3 h bins from 2000 to 2009 is not reflected in the performance of the T2m product (Figure 7).

Figure 9 shows T2m$_{sat}$-T2m$_{insitu}$ differences plotted as a function of AASTI L3 skin temperature for land ice and sea ice, respectively. Over land ice, the standard deviation decreases as a function of IST$_{skin\_L3}$, while the bias is around zero for IST$_{skin\_L3}$ between -45°C and -10°C, positive for higher temperatures and negative for lower temperatures. For sea ice, the maximum standard deviation is found at skin temperatures of about -20°C, with smaller standard deviations for higher and lower IST$_{skin\_L3}$. Positive biases are found for very cold skin temperatures (< -25°C) and for temperatures around the melting

point (> -4°C), while the intermediate temperatures have a slightly negative bias. This effect is included in the uncertainty estimates as presented in Sect. 3.2, which include IST$_{skin\_L3}$ as a predictor for both land ice and sea ice.

Figure 10 shows the validation results of the estimated uncertainties, where the T2m$_{sat}$-T2m$_{insitu}$ difference is plotted against the theoretical total uncertainties as obtained in Sect. 3.2 for land ice and sea ice, respectively. The dashed lines represent the ideal uncertainty with the assumptions that the in situ observations have an uncertainty of 0.1°C and that the sampling

uncertainty is 0.5°C. The estimated uncertainties show good agreement with the observed uncertainties, when the error bars follow the dashed line, which is the case here for both land ice and sea ice.

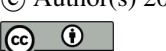



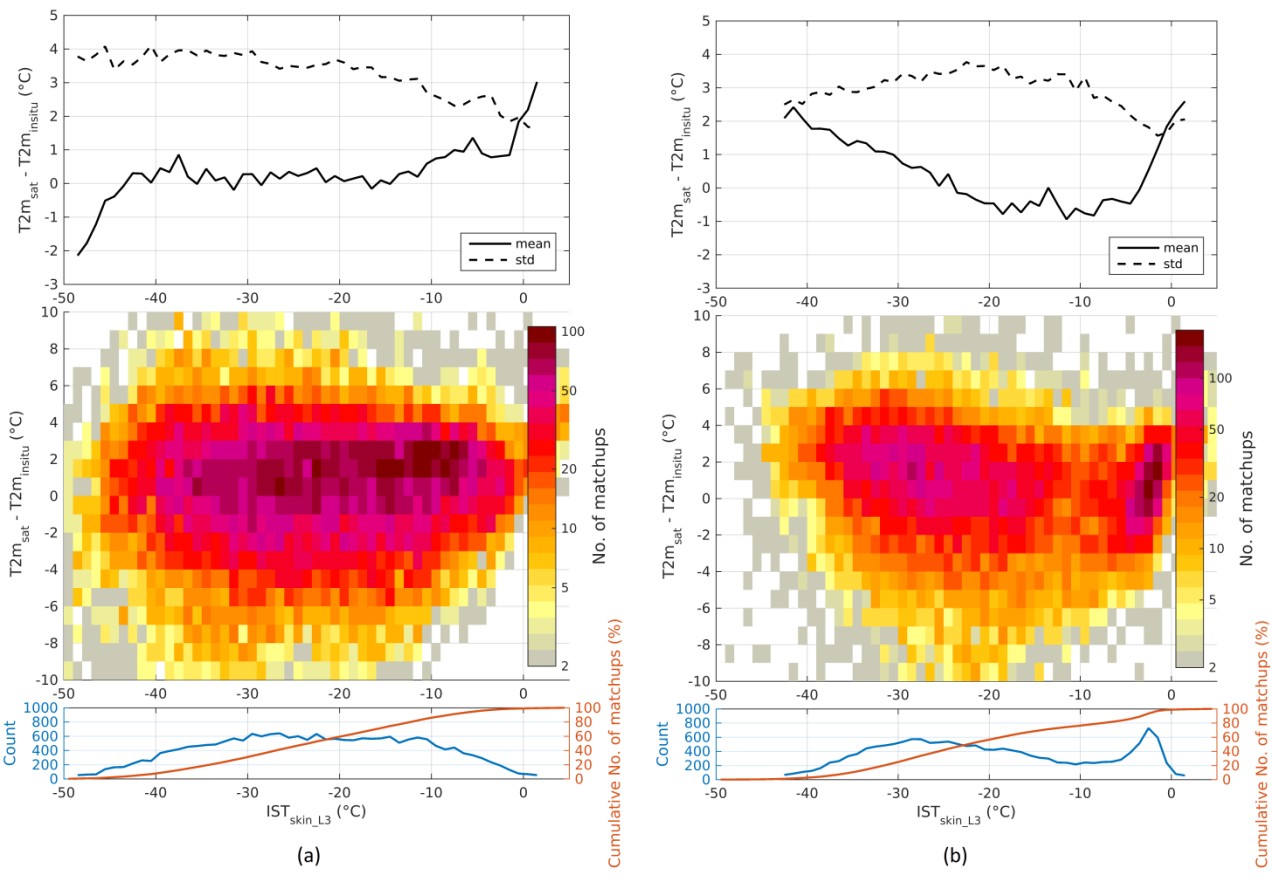

(a)                                                                           (b)

**Figure 9: Estimated T2m minus observed T2m (bin size of 1°C) as a function of binned (bin size of 1°C) satellite IST$_{skin\_L3}$ for (a) land ice and (b) sea ice. The dashed lines are standard deviations while the solid lines are bias in the upper figure. The surface plots in the middle figures show the number of matchups in each bin while the bottom plots show the number of matchups (blue) and the cumulative percentage of matchups (red) in each IST$_{skin\_L3}$ bin.**





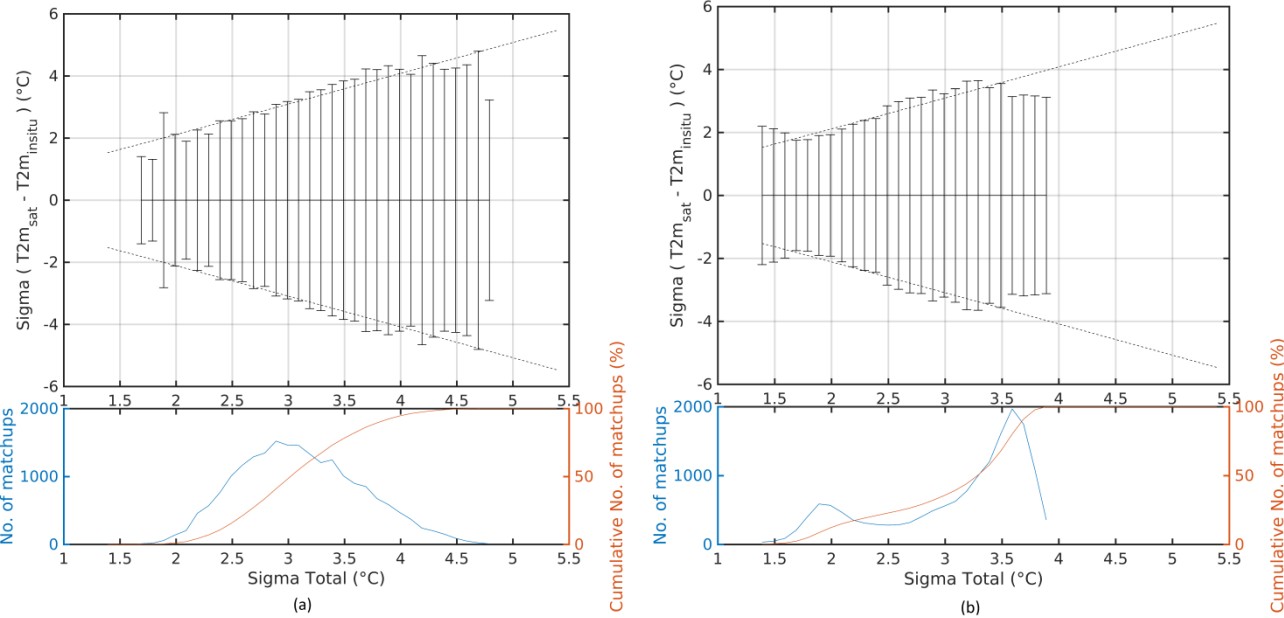

**Figure 10: Satellite estimated T2m uncertainty validation with respect to independent in situ T2m for (a) land ice and (b) sea ice. Dashed lines show the modelled uncertainty accounting for uncertainties in the in situ T2m and the sampling error. Solid black lines show one standard deviation of the estimated minus in situ differences for each 0.1 °C bin. The bottom plots show the number of matchups (blue) and the cumulative percentage of matchups for each bin (red).**

The performance of T2m$_{sat}$ has been compared to the performance of T2m from ECMWF's reanalysis ERA-I (T2m$_{ERA-I}$; Dee et al., 2011) and the replacement reanalysis, ERA5 (T2m$_{ERA5}$; Hersbach et al., 2020). Table 7 shows the performance of T2m$_{ERA-I}$ and T2m$_{ERA5}$ against the independent in situ T2m observations, which should be compared with the performance of the regression derived T2m$_{sat}$ as shown in Table 6. The comparison may not be truly independent as a number of stations and buoys have been assimilated into the ERA-I and ERA5 data products (Dee et al., 2011; Hersbach et al., 2020), which would favour the reanalysis products in the comparison. Yet, the bias is significantly lower for T2m$_{sat}$ than for both T2m$_{ERA-I}$ and T2m$_{ERA-5}$, while the other validation parameters are similar, with slightly better correlation and standard deviation, but slightly worse RMS results for T2m$_{ERA}$.

**Table 7: Statistics on the relation between ERA-I/ERA5 and in situ measured temperatures for independent test data. N: number of matchups used for validation, Corr: correlation, bias: T2m$_{ERA}$ – T2m$_{insitu}$ difference, Std: standard deviation, RMS: root mean square difference.**

|  | N |  | Corr (%) | Bias (°C) | Std (°C) | RMS (°C) |
|---|---|---|---|---|---|---|
| Land ice | 20872 | ERA-I | 96.4 | 3.41 | 3.18 | 4.66 |
|  |  | ERA5 | 97.1 | 2.03 | 3.08 | 3.69 |
| Sea ice | 16111 | ERA-I | 96.9 | 1.14 | 3.02 | 3.22 |



| | ERA5 | 95.7 | 2.19 | 3.67 | 4.27 |
|---|---|---|---|---|---|

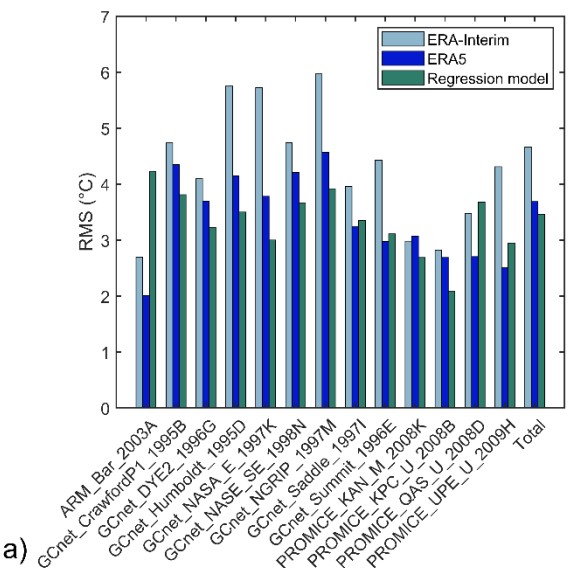

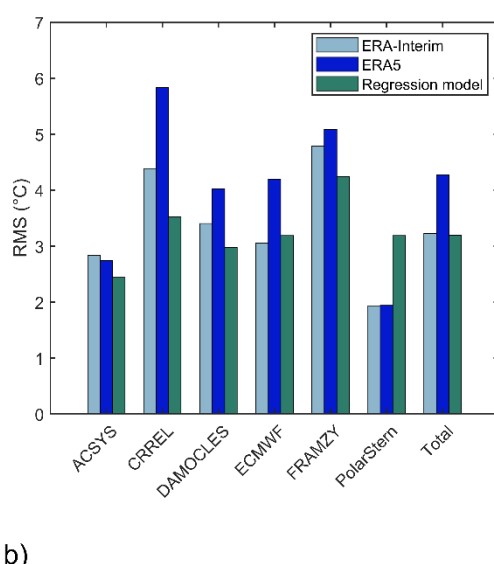

a)                                                                    b)

**Figure 11: Root mean square (RMS) differences calculated for the (a) land ice sites and (b) sea ice sites using T2m from ERA-Interim, ERA5 and the regression model, respectively. Only buoys with more than 200 observations are included. The last two bars listed as "total" are the RMS obtained by using all validation data.**

Figure 11 gives an indication of the performance of T2m$_{sat}$, T2m$_{ERA-I}$ and T2m$_{ERA5}$ at individual sites for each surface type. It shows the RMS difference between in situ measured T2m and T2m$_{ERA-I}$, T2m$_{ERA5}$ and T2m$_{sat}$, respectively, for the independent test sites and for both surface types. Due to the large number of buoys these have been validated for each data source with all observations weighted equally. The last bars refer to the RMS obtained by validating all test sites in one long time series weighting all daily observations equally. The total T2m$_{sat}$ agrees better with in situ observations for both surface types compared to both ERA-I and ERA5. ERA5 is significantly better than ERA-I over the GrIS, but ERA5 performs worse than both ERA-I and T2m$_{sat}$ over sea ice. Over sea ice T2m$_{ERA-I}$ agrees better with in situ observations from ECMWF data stream and Polarstern. However, these may be assimilated into both ERA-I and ERA5. The independent in situ observations by ACSYS, CRREL, DAMOCLES and FRAMZY are better reproduced by the satellite-derived T2m. The errors in the T2m$_{ERA-I}$/T2m$_{ERA5}$ and T2m$_{sat}$ data sets are expected to be independent and uncorrelated and a combination of either T2m$_{ERA-I}$ or T2m$_{ERA5}$ and T2m$_{sat}$ can therefore lead to an improved T2m estimate.

The monthly mean near surface air temperature estimates averaged over the GrIS have been shown in Fig. 12 for 2000-2009. The GrIS records a distinct annual cycle in near surface air temperature. The monthly mean air temperature typically reaches a maximum of -4°C during July and a minimum of about -28°C during winter. As is common for the Arctic environment, the temporal variability is largest during winter due to a more vigorous atmospheric circulation (Steffen, 1995).

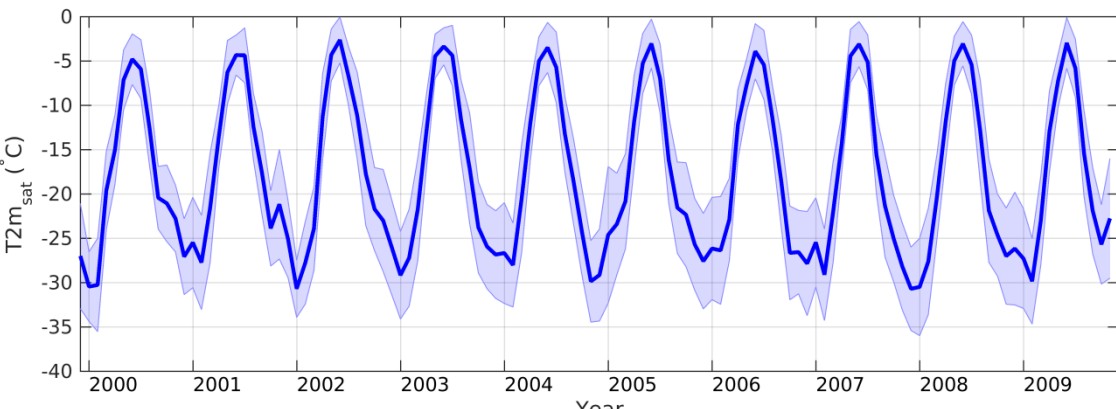

**Figure 12: Monthly mean T2m$_{sat}$ for the Greenland Ice Sheet. The shading represents the variability.**

The monthly mean T2m$_{sat}$ is shown in Fig. 13 for March, June, September and December averaged over the period 2000-

5    2009. The interior and northern part of the GrIS is typically colder than other parts of the Arctic in all months, while the

warmest regions are found along the sea ice marginal ice zone and the ablation zone of the GrIS. During summer little spatial

variability in monthly mean T2m is found over the Arctic sea ice.

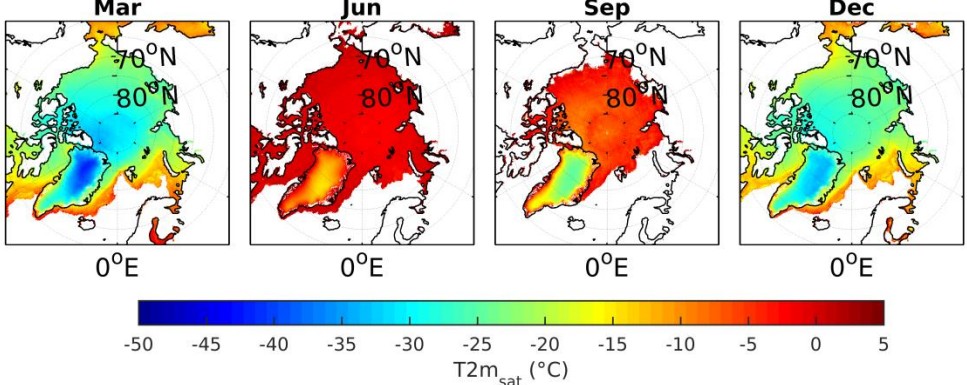

**Figure 13: Monthly mean T2m$_{sat}$ during March, June, September and December, averaged for the period 2000-2009.**

10  **5 Discussion**

Due to the limited number of in situ observations in the Arctic, and especially over sea ice, it is not a simple task to gather in

situ observations for testing and validating the regression models. The sparse number of in situ observations over sea ice is

the greatest challenge, as also discussed in Nielsen-Englyst et al., 2019. The lack of observations that represent all conditions

and regions in the Arctic and the resulting matching threshold of 15 km, combined with the large topographical variations

15   over the GrIS, inflate the uncertainty in the pixel-to-point comparison and will therefore complicate the derivation and

validation of the regression models.



The effects from topography over the GrIS have been assessed by introducing a new matchup dataset that ensures that the elevation difference between satellite and in-situ observations is less than 100 m over the GrIS. Excluding those AWSs (4 out of 23) with larger elevation difference than 100 m results in a reduction of the training data set of 2,935 matchups (i.e. from GC-net_JAR1 and PROMICE TAS_U) and a reduction in the validation data set of 560 matchups (i.e. from PROMICE

QAS_U and UPE_U). The performance of the satellite derived T2m improves, in particular the bias, which decreases to 0.07ºC, while the standard deviation decreases to 3.41ºC over land ice. ERA-I and ERA5 show limited changes in performance, with slightly increased biases of 3.48ºC and 2.07ºC, and standard deviations of 3.14ºC and 3.08ºC, respectively, when introducing the new matchup dataset over land ice. A similar good performance of the regression model is found when the 2 AWSs in the validation subset is kept. Despite the increased performance of the regression model, we

have included all observations in the training of the model to ensure a robust and spatial representative solution.

As infrared satellites cannot measure the surface temperature during cloudy conditions a cold clear-sky bias is often observed in infrared satellite $IST_{skin\_L3}$ averages compared to all-sky temperature averages (see e.g. Table 2). When using satellite $IST_{skin\_L3}$ observations it is thus important to assess the clear-sky bias, which varies with different temporal averaging windows (Nielsen-Englyst et al., 2019). However, through the use of an empirical statistical method, which is

trained against daily averaged in situ 2 m air temperatures (obtained both in clear sky and cloudy conditions), the conversion from $IST_{skin\_L3}$ to $T2m_{sat}$ removes the systematic $IST_{skin\_L3}$ clear-sky bias effects that may be present in the satellite data. As a result, we obtain a $T2m_{sat}$ estimate, which performs similar or better than the $IST_{skin\_L3,}$ when compared against in situ observations.

For short-lasting (<24 hours) cloudy conditions, the division into 3 h bin averages and the requirement of filled 3 h bins both

during night (between 18 and 6 local solar time) and day (between 6 and 18 local solar time) ensure that the diurnal cycle is best resolved despite the gaps with clouds. For long-lasting (>= 24 hours) cloudy conditions $IST_{skin\_L3}$ is not available and we do not retrieve $T2m_{sat}$ for these days. A statistical technique or the use of atmospheric models and assimilation may be used to fill in the gaps. By using a statistical model to combine in situ observed and clear sky satellite derived T2m estimates (over land, lakes, ocean and ice), including uncertainty estimates, EUSTACE has provided a global and gap free daily

analysis of surface air temperatures from 1850 to 2015 (Morice et al., 2019; Rayner et al., 2020).

The product derived here provides an increasing coverage over the time period from 2000-2003 and a stable coverage for 2003-2009. The average daily coverage is 84% and 67% for land ice and sea ice, respectively, considering the stable 2003-2009 period and the 0.25 degree grid. When considering a 1 degree grid resolution, these numbers increase to 94% and 81%, respectively. The high percentages in coverage demonstrate that the gaps due to cloudy days are limited and that the data set

contains a significant amount of information on the all-sky daily T2m even though it is based upon clear sky satellite observations.

Previous studies show a strong dependence of wind speed for both land ice and sea ice, but with different dependencies (Adolph et al., 2018; Hudson and Brandt, 2005; Miller et al., 2013; Nielsen-Englyst et al., 2019). However, the performance of the satellite derived T2m product did not improve much, when including the wind speed information from ERA-I or



ERA5 (Table 3). The reason is that the quality of the wind speeds is not adequate for use in the relationship model. Especially, the representation of katabatic winds in numerical weather prediction (NWP) models is a challenging task due to a high resolution needed in the vertical (Grisogono et al., 2007; Steeneveld, 2014; Weng and Taylor, 2003; Zilitinkevich et al., 2006), but also the processes of snow surface coupling, radiation and turbulent mixing are hampered by limited resolution, while their relative importance varies with wind speed (Sterk et al., 2013). More accurate information on the wind speed would likely improve the performance of the regression model when including wind speed as predictor. In particular, the higher resolution NWP output may be very beneficial in the regions of the GrIS, where the local topography interacts with the wind through katabatic effects (DuVivier and Cassano, 2013; Oltmanns et al., 2015; Renfrew, 2004). Regional high resolution reanalysis products are currently being carried out within the Copernicus Arctic regional Reanalysis service C3S project (https://climate.copernicus.eu/copernicus-arctic-regional-reanalysis-service). It is likely that such products will provide winds that can be used within a relationship model.

The T2m$_{SAT}$ data set developed here only covers the Arctic, but the AASTI data set also covers the Antarctica. This implies that similar statistical methods can be derived for the Antarctic ice cap and sea ice. Preliminary investigations indicate that a T2m product can be derived for the ice caps with similar performance as for the GrIS, whereas the Southern Ocean sea ice is challenging due to very few in situ observations (Morice et al., 2012). For both Southern regions, more in situ observations are needed to repeat the work performed for the Arctic and to determine a reliable statistical model.

Including other available satellite products, such as Modis IST observations (Hall et al., 2004) or the (A)ATSR data set (Ghent et al., 2017) could improve the quality of the T2m$_{SAT}$ product. However, adding new data requires a detailed knowledge of the characteristics of the data set, such as sampling frequency and uncertainty of the IST observations. In addition, determination of the relationship model is needed again. At the same time, adding more satellite overpasses to the daily estimates may not improve the uncertainty of the products. This is evident when comparing Fig. 7 and 8 where the variation in the number of satellite observations during the record (Fig. 8) is not reflected in a similar variation in the performance of the product (Fig. 7). The uncertainty in the beginning of the record is comparable to the uncertainty in the end of the record, despite an almost doubling of the observed 3 hourly averages throughout the day.

The AASTI version builds upon the Clara version 1 data set from the CM-SAF. A version 2 of the data set is now available (Karlsson et al., 2017), facilitating the production of an AASTI version 2 data set that covers from 1982 up to present. With a consistency in the retrieval algorithm and data sets, it will be possible to use the relationship model to produce a satellite based climate data record of T2m from 1982 to today.

## 6 Conclusions

The air temperature over land ice and sea ice is an obvious indicator for Arctic climate change and it can easily be compared with climate change indicators from other regions. This study introduces a methodology for using satellite skin temperatures for estimating air temperatures, to compensate for the lack of in situ measurements, and as a supplement to reanalysis





products. Daily near surface air temperatures (T2m) have been estimated based on daily clear sky satellite Level 3 (L3) observations of ice surface skin temperatures ($IST_{skin\_L3}$) in the Arctic, using the Arctic and Antarctic ice Surface Temperatures from thermal Infrared satellite sensors (AASTI) reanalysis. A regression based method has been used and tuned against in situ observed T2m using $IST_{skin\_L3}$ observations covering both Arctic sea ice and land ice. In general, there is

a good correlation between T2m and $IST_{skin\_L3}$, due to the seasonal cycle in both IST and T2m. Different predictors have been tested to examine how to best capture the variability in the T2m-IST difference. These predictors include latitude, theoretical downward shortwave radiation (not considering clouds), seasonal cycle, and wind speed (ERA-I and ERA5) and were selected based on the current knowledge from the literature (Adolph et al., 2018; Hall et al., 2008; Hudson and Brandt, 2005; Nielsen-Englyst et al., 2019; Vihma and Pirazzini, 2005), limited by the available data. The seasonal cycle was

introduced based upon the results from an analysis of in situ observations, where a seasonal cycle in the relationship between surface skin and near surface temperature was observed (Nielsen-Englyst et al., 2019). The highest correlation and lowest RMS against the training data was found using a model where $T2m_{sat}$ is predicted from daily satellite $IST_{skin\_L3}$ combined with a seasonal variation, assumed to have the shape of an annual harmonic. This model has been used to derive daily T2m on a 0.25 degree regular latitude-longitude grid from the clear sky AASTI $IST_{skin\_L3}$ over the Arctic during the period 2000-

2009 (Kennedy et al., 2019), where different regression coefficients have been used for land ice and sea ice. Days with clouds or limited clear sky observations have been excluded from the analysis. Considering a 1 degree regular latitude-longitude grid, the average daily coverage of the $T2m_{sat}$ product is 94% over the GrIS and 81% for sea ice, considering the years 2003-2009. The days when the $T2m_{sat}$ is available, the T2m estimate can be considered as a daily averaged all-sky T2m, since it has been tuned against all-sky in situ observations.

The estimated $T2m_{sat}$ data record has been validated against independent in situ measured 2 m air temperatures. The validation results indicate average biases of 0.30°C and 0.35°C and average root mean square errors of 3.47°C and 3.20°C for land ice and sea ice, respectively. An uncertainty model has been developed and all daily $T2m_{sat}$ estimates come with a total uncertainty divided into a random, locally systematic and large-scale systematic uncertainty component. The total uncertainty of the satellite derived $T2m_{sat}$ shows good validation results when validated against independent in situ

observations.

The satellite derived $T2m_{sat}$ product has been compared to ERA-I and ERA5 T2m estimates and has proven to validate similar or better than both of these. The $T2m_{sat}$ product is independent of the quality of the NWP forecasts and thus represents an important supplement to the model based T2m. The errors in NWP products (e.g. $T2m_{ERA-I}$ or $T2m_{ERA5}$) and the errors in the product derived here ($T2m_{sat}$) are expected to be independent and uncorrelated and a combination of a NWP

product and the $T2m_{sat}$ data can therefore lead to an even better T2m estimate. The regression models presented here both work on satellite observations that are available from reprocessed records but opens up for a near real time estimation of T2m from satellites. The results obtained for the ice covered areas show that there is a large potential for using satellite observed surface temperatures to estimate near surface air temperatures. However, these estimates are not supposed to




replace the already existing air temperature measurements, but rather to supplement these e.g. in areas where no in situ observations are currently available.

## 7. Data availability

The derived surface air temperatures from satellite surface skin temperatures over ice can be downloaded from

http://dx.doi.org/10.5285/f883e197594f4fbaae6edebafb3fddb3 (Kennedy et al., 2019). The PROMICE data can be accessed through http://www.promice.dk (last access: 16 November 2018). The ARM data are available at https://www.archive.arm.gov/discovery/\#v/results/s/s::co (last access: 21 December 2018). GC-Net data can be found through doi:10.5067/6S7UHUH2K5RI  (Kindig, 2010). Data from CRREL mass balance buoys are available from: http://imb-crrel-dartmouth.org (last access: 24 November 2016), while POLARSTERN data can be downloaded at

https://dship.awi.de/Polarstern.html (last access: 24 November 2016. FRAMZY data are available from doi:10.1594/WDCC/UNI_HH_MI_FRAMZY2002 (Brümmer et al., 2012b), http://dx.doi.org/doi:10.1594/WDCC/UNI_HH_MI_FRAMZY2007 (Brümmer et al., 2011b), and http://dx.doi.org/doi:10.1594/WDCC/UNI_HH_MI_FRAMZY2008 (Brümmer et al., 2011c), while ACSYS data are found here: DOI: 10.1594/WDCC/UNI_HH_MI_ACSYS2003. Damocles data can be found here:

doi:10.1594/wdcc/uni_HH_MI_DAMOCLES2007 (Brümmer et al., 2011a). The traditional buoy and ship data obtained from ECMWF are distributed through the World Meteorological Organization's (WMO) Global Telecommunication System (GTS) and available for members at the ECMWF Meteorological Archival and Retrieval System (MARS). Finally, the AASTI $IST_{skin\_L2}$ data are available from http://dx.doi.org/10.5285/60b820fa10804fca9c3f1ddfa5ef42a1 (Høyer et al., 2019).

## 8 Author contribution

Pia Nielsen-Englyst, Kristine S. Madsen and Gorm Dybkjær compiled and quality checked the in situ data. Pia Nielsen-Englyst, Jacob L. Høyer and Kristine S. Madsen designed and developed the regression model and estimated uncertainties. Gorm Dybkjær, Jacob L. Høyer and Rasmus Tonboe developed the AASTI $IST_{skin\_L2}$ data. Sotirios Skarpalezos did the ERA5 matchup. Pia Nielsen-Englyst prepared the manuscript with contributions from all authors.

## 9 Competing interests

The authors declare that they have no conflict of interest.



## 10 Acknowledgements

This study was carried out as a part of the European Union Surface Temperatures for All Corners of Earth (EUSTACE), which is financed by the European Union's Horizon 2020 Programme for Research and Innovation, under Grant Agreement no 640171. The aim of EUSTACE is to provide a spatially complete daily field of air temperatures since 1850 by combining
satellite and in situ observations. The author would also like to thank the data providers.

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
