# Peer review of "Deriving Arctic 2 m air temperatures over snow and ice from satellite surface temperature measurements"

_The Cryosphere, 2021_

## Author Comment (AC1)

**Referee #1:**

**General Comments**

Overall, this interesting and valuable paper is generally well written. There is some additional discussion needed in places and some tightening of prose, particularly in the second half of the introduction and in the discussion/conclusion. Yet the research itself is comprehensive and the science is good.

**Specific Comments**

- After Page 2 Line 30 the introduction needs a bit of attention to retain the good quality of previous paragraphs. It felt like the flow of the narrative was lost around here and there is a bit of repetition.
  The second part of the introduction has been rewritten to retain the quality of the first part.
- For the satellite data please give an explanation for the choice of a 0.25 by 0.25 degrees regular geographical grid. This seems quite a coarse resolution given the input data, is this to correspond to T2m datasets from other sources or model grids?
  The 0.25 degree grid was chosen within the EUSTACE project to ensure a common grid that could be used globally and that covers the surface skin temperature for all surfaces: ocean, land, lake and ice. The final products within EUSTACE was on this 0.25 deg grid. A sentence on this has been added to the manuscript.
- Page 7 Lines 5-7:
  - It took a couple of reads to figure out that the ISTskin_L3 is the daily version of the L3 not the 3 hourly (assuming I understand correctly). I think changing the text to something like "Here, we have aggregated the AASTI ISTskin_L2 observations into 3 hourly and daily, gridded 5 Level 3 (L3) averages of ISTskin_L2 on a fixed 0.25 by 0.25 degrees regular geographical grid. The daily gridded averages (ISTskin_L3) are calculated by averaging" might make this clearer. The text has been changed as suggested.
- Page 8 Line 26:
  - You mention the distances but not temporal matching. I guess from the data this is a daily average comparison, but this might be worth mentioning explicitly here.
    It has now been stated that is a daily average comparison.
- Page 14 28-29:
  - This sentence reads as though only the random uncertainties are provided for each pixel and the others are provided as one value for the whole dataset. Is this correct?
    The sentence has been reformulated. All uncertainty components are provided for each pixel as explained in Section 3.2. Thanks for pointing this out.
  - Surely given the inclusion of a sampling uncertainty in the synoptic component and separate equations for land ice and sea ice there should be a value for each pixel? This is the case for synoptic uncertainties for similar satellite products such as from e.g. Ghent et al (2017).
    Agree. See above.

- Page 15 Line 1-4:
  - This sentence makes it sound like only ice over sea is included; ice shelf (land ice) and sea ice. But Figure 6 and previous figures show data over the Greenland ice sheet. Is the Greenland ice sheet included in the dataset? Yes, the Greenland Ice Sheet is included. The sentence has been reformulated.
- Results:
  - I would like to see a few more citations and additional discussion of the results here as I was often left wondering how the results compared with previous research or observations that might back up your results. For example:
    - "The monthly mean air temperature typically reaches 20 a maximum of -4°C during July and a minimum of about -28°C during winter." Does this correspond to previous research and/or observations? Perhaps the GrIS in situ observations could be added to Figure 12 to show how close the T2m_sat is? The range in monthly mean air temperature is in agreement with those reported by van As et al. (2011) at a number of PROMICE AWSs. This has now been stated in the manuscript. In relation to Fig 12, we don't think it will be a good idea to include in situ observations here. Fig. 12 shows the average temperature of the entire GrIS, and we don't think the average of the sparse in situ observations can provide a spatially representative measure of the mean T2m of Greenland for the period.

    - It seems that the results from comparisons of ERA-Interim (and ERA-5) suggest that the long standing warm bias in these reanalyses for the Arctic still exists, which deserves some comment here and perhaps citation of previous studies on this. I also don't know if ERA-5 has been compared to in situ T2m over the Arctic in other research yet so this could be an interesting result given the warm bias may still be a feature of this dataset series. Again, if so this is worth noting. Thanks for pointing this out. Previous studies have been cited and two recent studies (Wang et al. 2019 and Graham et al. 2019) have evaluated both ERA-I and ERA5 for Arctic sea ice against buoy observations and N-ICE2015, respectively, and found that ERA5 also suffers from a warm bias. Similarly, recent studies found no significant improvement in performance over the GrIS for ERA5 compared to ERA-I (Delhasse et al., 2020; Zhang et al., 2021). This has now been stated in manuscript.

    - Page 17 Lines 5-9, I believe your statements about Arctic cloud cover are correct, but I think a citation would be useful to back this statement up. Citations have been added.
  - Discussion:

- This section was more like a list of additional things to note that did not fit in the rest of the paper rather than a coherent discussion. I felt it didn't really tell the story of the research in the way it deserves. The section has been restructured and rewritten to make it coherent and to include more references to the result section and previous research.
- I think this needs a bit of work to restructure and possibly a short summary of results relevant to each point made. For example:
  - Page 22 Lines 11-18 seem to refer to the fact that "The satellite 10 derived air temperatures are about 0.3°C warmer than measured in situ air temperature for both land ice and sea ice." which is probably due to the influence of the linear regression? It would be nice to include the context for these sorts of statements in the discussion. These lines actually refer to the cold sky bias observed when satellite L3 IST is compared with in situ observations (~-2°C as shown in table 2). The section has been rewritten and hopefully it is more clear now what we have done to remove this clear-sky bias in the final product (for which the bias is 0.3°C for both surfaces). As shown in Sect. 4.3, part of the remaining bias for the GrIS is likely explained by topographic effects.
- Will the surface temperature dataset be extended to seasonal snow and ice? It is possible to extend this work to seasonal snow. It requires a dynamical surface mask and repeating the derivation of a regression model. However, within EUSTACE similar efforts have already been made to cover seasonal snow (Morice et al., 2019; Rayner et al., 2020; Good 2015). This point has been added to the discussion section.

**Technical Corrections**
- Page 2 Line 19: Satellite not Satellites. Accepted
- Page 2 Line 23: Either un-capitalise The or remove. Removed
- Figure 3, 4: might be a big ask depending on the software used, but is there any chance of removing the sliver of white from the plots? The figures have been updated and the silver of white has been removed.
- Page 8 Line 22: I think a subheading for this validation section would be useful to the reader. Accepted
- Page 16 Line 8: there is not there are. Accepted
- Page 23 Line 17: MODIS not Modis (acronym of sensor). Accepted
- Page 15 Line 2: could you expand the acronym ETOPO1 or provide a very brief indication in text of what this is? Perhaps "ETOPO1 global relief model" or similar? Accepted
- Conclusion section: a bit long and should be distilled down to the major points. The section has been shortened a bit, and now excludes the listing of tested models that where not used.

---

## Author Comment (AC2)

**Referee #2:**

General Comments

This is an exceptionally well-written contribution; if only all papers were so clear. The datasets collected seem very comprehensive, and the careful attention to uncertainties in satellite estimates is a strength, as is the good practice of independent validation of both the data and the data uncertainties. The value of this observational effort (compared to just taking re-analysis outputs) is demonstrated. The conclusions are well supported by the discussion.

Specific Issues

8.5 - Why should surface melting situations necessarily be excluded? How common is the situation of the (wet) skin temperature being > 5 deg compared to it being a wrong observation? I guess this is explained more in the reference, but perhaps a little more comment here would be justified.

The skin temperatures warmer than 5 deg are usually located along the coasts and sea ice edge, and not representing warm (wet/melting) ice, but indicating inconsistency between the ice mask used and the surface temperatures. The sea ice mask is based on the coarser resolution OSI-SAF product, which is subject to land-spillover effects (see left figure below). Better spatial resolution (as for SICCI-25km) reduces the spurious ice along the coast (see figure to the right). Here, the warm (>5deg) surface temperatures have been used as a filter to minimize the cases of spurious ice. This sentence of the manuscript has now been reformulated and the above explanation has been added to the manuscript as well. Thank for pointing this out.

[Figure]

**Number of observations with coincident CCI SST > 3°C and SIC>15 % for OSISAF (left) and SICCI-25km (right) during 2009.**

Figure 4 includes open ocean areas, so this presumably is the SD of any surface present not of specifically ice surfaces, despite the wording of 8.12? Yes, this is correct and the sentence has been reformulated.

"Daily mean" surface air temperature data from weather stations are often actually the mean of the daily max and daily min reported. Is that the definition applied to the in situ data in section 3.1 here? No, here the "daily mean" is actually the average of all observations available for the given day (e.g. the hourly observations provided by PROMICE). This has been stated in 4.30.

Technical / editorial

1.10 - unnecessary hyphen after weather. Deleted

1.11 - meter -> metre and also throughout eg 8.25 etc; reserve "meter" for an instrument. Accepted

2.23 - micron -> micrometre. Accepted

Figure 4 caption: not totally unambiguous what calculation this is, but I think it is the SD for the named month of each year, then averaged over years? Yes, this is correct and the caption has been updated.

12.3 -- explain "theoretical shortwave radiation" -- is this top of atmosphere to give seasonality?

Yes, it is the theoretical top of atmosphere shortwave radiation. This has now been stated in manuscript.

---

## Author Comment (AC3)

**Referee #3:**

The manuscript addresses estimation of 2-m air temperature (T2m) over the Greenland Ice sheet and the Arctic Ocean sea ice. The work is based on satellite remote sensing data on clear-sky ice/snow surface temperatures and their comparison with in-situ observations on T2m. I am impressed by the amount of data analysed, originating from different sources, and by the carefulness and detail of the analyses made. Among others, a lot of attention is paid on uncertainty analyses. The manuscript is well organized, the text is well written, and the illustrations are of high quality. The results are important, convincingly demonstrating that T2m estimates based on satellite remote sensing data are useful and reasonably accurate. Hence, they can supplement reanalysis products and rare in-situ observations on T2m. I suggest accepting the manuscript subject to minor revisions.

Specific comments

I wish to see more attention to the seasonal distribution and accuracy of the satellite-based T2m estimates. The quantitative results are given for the entire data set (e.g. page 22, lines 26-28), but it is well known that over the Arctic sea ice clouds are more common in summer than other seasons. Figure 8 demonstrates that there are less data from summers, but Figure 7 suggests that the estimates are better in summer than in winter, which must be related to smaller thermal differences between the ice and air in summer. Detailed comparison of skill scores (bias and rmse) between different seasons is, however, missing, and it remains unclear how useful the satellite-based T2m product is over sea ice in summer. This could be easily improved by presenting quantitative results for different seasons.

Thanks for pointing this out. We have added a figure (Fig 7; shown below) to the manuscript, which shows the seasonal variation in bias, standard deviation and the number of matchups. We agree that the thermal difference is smaller during summer, but this to first order, should be captured by the seasonal component of the regression model. The smaller stddev during summer is likely a combination of better performance due to improved cloud screening during summer and the smaller natural variability that is observed in summer conditions.

[Figure]

We have also added a figure (Fig. 8) to illustrate the coverage of the final T2m$_{sat}$ product in the end of the discussion section. As expected we see a minimum in coverage over sea ice in summer due to an increased cloud cover.

[Figure]

[Figure]

It would be good to make it clear already in the Abstract that the study addresses the Arctic Ocean and Greenland Ice Sheet, but not other land ice in the Arctic. Related to this, in Table 1 land snow must refer to seasonal snow. However, in the Discussion, Conclusions, and Abstract, there is nothing mentioned about results for seasonal snow. Were the results similar to those for land ice?

Yes, we agree on this. It has now been stated already in the abstract as well as in the introduction and conclusion. The study also includes the two ARM stations (BAR + ATQ) with seasonal snow cover. These are included as land ice stations in this study and only data from snow covered periods have been used. This is not very clear from the original manuscript, which has now been updated (see Sect 2.1). ATQ was used for training while BAR was used for validation. The RMS error at Barrows is shown in figure 12 (very first bar). From this figure, the results look similar to those for land ice. The T2m$_{sat}$ validation against Barrows gave a mean correlation of 95.8%, a bias of 2.49°C and std of 3.42°C. The correlation and std are similar to the average values over land ice while the bias is considerably larger. This is likely explained by physical differences between seasonal snow covered sites and GrIS sites, which are not fully captured by the regression model. This has now been stated in manuscript Sect. 4.2. In the discussion section, we have added a sentence on the possible extensions of this product to seasonal snow, which will require a dynamical surface mask and repeating the derivation of a regression model.

Page 1, last sentence: Could the information be updated (2010s vs. 2000s) and extended to also address the strong winter warming over the Arctic sea ice? The latter is evident on the basis of reanalyses.

Yes, the information has been updated according to the new historical climate data collection from DMI (Cappelen ed., 2021) and the winter warming over the Arctic has been addressed in the very first sentence.

Page 2, lines 7-8: "In particular" is questionable. It is even more difficult to achieve climate-quality precipitation and air humidity records from the region.

We agree on this and have replaced "In particular" with "Therefore".

Page 3, lines 4-5: Make it clear that the zones refer to the Greenland ice sheet.

Accepted

Table 1: Use of Polarstern data as a proxy for 2-m air temperature sounds surprising. According to my knowledge, in the Polarstern weather station the air temperature sensors are located 29 m

above the sea surface. In Figure 11, however, the rms-errors are small for Polarstern. Could it be because the data are mostly from summer, when the vertical temperature gradients in the boundary layer are mostly small?

You are right. The Polarstern air temperatures are measured 29 m above the sea surface. These measurements are not used in the derivation of the regression model, but only for validation. As noted by the reviewer the validation against Polarstern is relatively good. We agree that this is most likely explained by the fact that data are mainly from summer, when the vertical temperature gradients are mostly small, and the performance of the cloud screening algorithm reaches its maximum. This has been stated in manuscript.

Page 20, line 21: The temporal variability of air temperature is largest in winter because the meridional temperature gradient is largest, the atmospheric circulation is vigorous, and the cloud radiative effect is large (compared to near-zero in summer).

The explanation has been extended to include all of above.

Page 22, lines 1-10: As new results are presented here, consider moving this paragraph to the Results section.

The section has been moved as suggested. Also, the result section has now been divided into subsections to improve readability.

End of page 22 – beginning of page 23: I am surprised that the use of wind speed did not improve the results over sea ice (over Greenland Ice Sheet, the reanalysis errors for near-surface winds may be so large that it is not so surprising there). Also, it looks strange that the correlation coefficients and RMS-errors are exactly the same for the four regressions presented for sea ice in Table 3. I suggest double-checking the calculations.

The calculated values have been double-checked and the RMS-errors are exactly the same when using 2-digits, but this is not the case when using 3 digits. We found it also surprising that the inclusion of wind speed did not improve the results over sea ice. Possible explanations are provided in the discussion section and include limitations in the representation of surface roughness and the processes of snow-surface coupling, radiation and turbulent mixing partly due to limited resolution and varying relative importance of the processes with wind speed (Sterk et al., 2013). Better quality of wind speed (e.g. from regional high resolution reanalyses such as CARRA) will likely improve the performance when included in the regression models.

Page 23, line 30: In summer, the air temperature over sea ice in not so good indicator for Arctic climate change. Ice and snow are rapidly declining, but T2m remains the same, close to 0 deg C, as long as there is sea ice left. See e.g. Figure 2g in Vihma et al. (2008) (just for information, no reason to cite).

We agree on this and the sentence has been reformulated.

Page 24, last paragraph: The independence of the satellite-based T2m product and reanalyses is stressed here but, as mentioned earlier, reanalyses actively assimilate available in-situ observations. In particular, the ECMWF has had a pioneering role in assimilation of T2m data, which was done also in the Arctic already in the old ERA-40 reanalysis. However, I fully agree that the satellite-based T2m product is very valuable, and may indeed yield even better results than assimilation of (rare) observations into a reanalysis system. T2m is a diagnostic product in reanalyses, calculated on the basis of surface temperature and air temperature at the lowest model level, and liable to uncertainties in the parameterization of turbulence in the stably stratified atmospheric boundary layer. Note, however, that the NCEP-CFSR reanalysis, which is based on a

coupled atmosphere – sea ice – ocean model, has performed better for near-surface atmospheric variables over sea ice.

We are not entirely sure what the reviewer is after in this comment. We agree with what the reviewer states about the assimilation of T2m into the models and it has now been stressed here as well for clarification. We see the derived product as a supplemental data set to the in situ observations in the data assimilation and this has now been clarified in the very last sentence. Moreover, we have added a comment on the better performance of NCEP-CFSR in Sect. 4.2.

Figure 6: In the legend, replace "air surface temperature" by "near-surface air temperature" or "2-m air temperature".

Accepted